# Impact of spinal fusion on severity health status in scoliotic adolescents with polyhandicap

Hugo Bessaguet[1,2]*, Marie-Christine Rousseau[3,4], Vincent Gautheron[1,2], Etienne Ojardias[1,5], Bruno Dohin[2,6]

1 Department of Physical and Rehabilitation Medicine, University Hospital of Saint-Etienne, Saint-Etienne, France, 2 Saint-Etienne University Jean Monnet, Lyon 1, Savoie Mont-Blanc University, Laboratoire Interuniversitaire de Biologie de la Motricité, Saint Etienne, France, 3 EA 3279, CEReSS, Research Centre on Health Services and Quality of Life, Aix Marseille University, Marseille, France, 4 Fédération des Hôpitaux de Polyhandicap et Multihandicap, San Salvadour Hospital, University Hospital of Paris, Hyères, France, 5 Lyon Neuroscience Research Center, Trajectoires Team, Inserm UMR-S 1028, CNRS UMR 5292, Lyon1 & Saint-Etienne Universities, Saint-Étienne, France, 6 Department of Pediatric Orthopedic Surgery, University Hospital of Saint-Etienne, Saint-Priest-en-Jarez, France

* hugo.bessaguet@univ-st-etienne.fr

**Data Availability Statement:** All relevant data are within the manuscript and its Supporting information files.

## Abstract

### Background

Scoliosis constitutes a prevalent comorbidity in adolescents with polyhandicap and frequently leads to other severe impairments, impacting abilities and requiring complex caregiving strategies. Therefore, spinal fusion surgeries are commonly performed to alleviate pain and provide more comfort. However, spine stabilization has not previously been proven to improve the severity health status of adolescents with polyhandicap according to specific clinical scales.

### Objective

This study describes and compares the severity health status of adolescents with polyhandicap before and after they underwent spinal fusion.

### Methods

A monocentric retrospective observational study was conducted in the university hospital centre of Saint-Etienne, France. We included between 2009 to 2020, 30 scoliotic adolescents with polyhandicap who underwent spinal fusion performed with the same surgical technique and the same surgeon. The main outcome was the variation in the Polyhandicap Severity Scale (PSS) score after surgery. Secondary outcomes were variations in PSS subscores, quality of life scores, fronto-sagittal X-ray parameters, and measures of surgical complication rates and lengths of stay.

### Results

Among 30 adolescents, 27 PSS analyses were performed. We found a significant improvement between pre- and postoperative PSS scores, mainly for pain and respiratory,

**Funding:** This research did not receive any specific grant from funding agencies in the public, commercial or not-for-profit sectors.

**Competing interests:** The authors have declared that no competing interests exist.

**Abbreviations:** abPSS, Polyhandicap Severity Scale abilities subscore; BMI, Body Mass Index; coPSS, Polyhandicap Severity Scale comorbidities and impairments subscore; CP, Cerebral Palsy; DWI, Deep Wound Infection; FB, Frontal Balance; FCA, Frontal Cobb Angle; FIM, Functional Independence Measure; GMFCS, Gross Motor Function Classification System; ICF, International Classification of Functioning, disability and health; ICU, Intensive Care Unit; IT, Intra Thecal; mCDS, modified Clavien-Dindo-Sink; OBL, ; PI, Pelvic Incidence; PLH, Polyhandicap; PolyQoL, Quality of life questionnaire for persons with polyhandicap; PSS, Polyhandicap Severity Scale; PT, Pelvic Til; QoL, Quality of life; SB, Sagittal Balance; SCA, Sagittal Cobb Angle; SS, Sacral Slope.

digestive, and skin disabilities. These improvements were accompanied by significant reductions in pelvic obliquity, in frontal and sagittal curves. The mean hospital length of stay was 45 days. During postoperative period, patients received a personalized postoperative rehabilitation procedure with spasticity and pain treatments, physiotherapy, and verticalization (wheelchair sitting and positioning devices such as contoured seat intended to increase postural stability). The mortality rate was estimated at 7%. At least 1 complication per patient occurred.

## Conclusions

We show that spinal fusion surgeries confer a significant improvement in the severity health status in scoliotic adolescents with polyhandicap.

## Introduction

Polyhandicap (PLH) is currently defined as a complex disability condition corresponding to a chronic affliction occurring in an immature brain [1], leading to the combination of severe/profound mental retardation and serious motor impairment, resulting in an extreme restriction of autonomy and communication [2,3].

As comorbidities accumulate in patients with PLH throughout their lives, impairments, activity limitations and health-related quality of life (QoL) worsen [4]. Among comorbidities in patients with PLH, scoliosis represents one of the most prevalent conditions with chronic respiratory insufficiency [5–7], chronic digestive disorders, or epilepsy [8]. Indeed, Rousseau et al. [9] estimated that the prevalence of severe scoliosis (scoliosis with Cobb angle >50˚) prevalence was 60.2%, accounting for a large proportion of deaths induced by respiratory failure. The latter, represents the main cause of death in adults with PLH [9] (63.2%).

Even if the natural history [10], risk factors, treatments [11,12] and complications [13] of scoliosis with Cobb angle >50˚ are well known in nonambulant cerebral palsy (CP) patients (Gross Motor Function Classification System—GMFCS level 5), few data are available in PLH [3].

Hodgkinson et al. [14] found severe neuro-orthopaedic impairments in adolescents with PLH were associated with poor general condition. They hypothesized that multidisciplinary therapeutic strategies must include surgical scoliosis treatment to improve pain, comfort, and positioning. However, Cassidy et al. [15] found no differences in terms of pain, function, or time for daily care when comparing adolescents with PLH who underwent thoracolumbar spinal fusion with the nonoperated control group. De Lattre et al. [16] also failed to demonstrate improvements in health status between non operated and operated adolescents with PLH. They also reported a high rate of per- and postoperative complications (93,7%). McCarthy et al. [17] reported a rate of perioperative death up to 7% in nonambulant CP patients.

Previous studies estimated that 12% to 32% of subjects with PLH underwent scoliosis surgery [9]. Thus, there is still considerable interest in investigating whether scoliosis surgery truly confers measurable benefits on subsequent health and daily life comfort in adolescents with PLH. If it is hypothesized that surgeries lead to measurable clinical and radiological benefits, valid clinical evaluations are lacking.

The aims of this study were first to evaluate whether the health status variation in adolescents with PLH was improved after scoliosis surgery and second to evaluate the incidence of perioperative complications.

## Material and methods

### Study design

We conducted a monocentric retrospective observational study, reviewing charts of adolescents with PLH who underwent surgery for severe neurologic scoliosis. Recruitment occurred in paediatric orthopedic surgery and physical rehabilitation medicine departments of the University Hospital Centre of Saint-Etienne. This study, conducted from September 2020 to June 2021, was approved by the institutional review board (Institutional Review Board: IORG0007394—IRBN492021/CHUSTE–"Terre d'Ethique" Research Committee–Hospital Centre of Saint-Etienne). All familial caregivers received written information about the study, according to French law, for retrospective studies. A written consent was collected from all caregivers. STROBE guidelines were followed to report this study.

### Participants

All adolescents with PLH who underwent definitive spinal fusion surgery for scoliosis with Cobb angle >50˚ were eligible. Additionally, surgical indication was determined through a multidisciplinary evaluation, considering various parameters including respiratory and abdominal disorders, complications in wheelchair installation, and severe axial hypotonia. PLH was defined as a combination of cerebral lesions onseted under 3 years old and responsible for severe motor deficiency with restricted mobility (GMFCS V), associated with profound intellectual impairment, and daily life dependence (Functional Independence Measure—FIM < 55) [18]. A minimum of 12 months follow-up was required for inclusion and postoperative data collection, except in adolescents presenting lethal complications during procedure or in the immediate follow-up, who were also considered in the analysis. Patients were operated on by one surgeon to ensure population homogeneity. Surgery consisted in all patients in an hybrid instrumentation of the spine: patients in supine position; pelvic fixation to correct the pelvis obliquity; instrumentation of the lumbar area with pedicular screws; thoracic area with sublaminar ligaments and hook-claw at the upper part of the instrumentation; rods were self-bowing by surgeon during the procedure; all implants from Medtronic (Medicrea™, Rillieux la Pape 69140, France).

Postoperative care systematically included one night in ICU for monitoring. Pain medication started peroperatively with intrathecal injection of morphine (5 μg/kg). A peridural catheter was maintained during 48h to 5 days with continual perfusion of low dose ropivacaine 0.2% (related to patient weight), and if necessary, paracetamol, ketoprofen, diazepam and oral morphine (related to patient weight and respiratory status) could be used. Esomeprazole for gastric ulcer prevention, lactulose per oral for constipation prevention, and enoxaparin if necessary (related to risk factors) were also administered. Feeding started one day after spinal fusion, through various modalities depending on patients. Lower limbs mobilizations were started two days postoperatively in sitting position, then performed every day. Analysis of targeted medical, paramedical records and X-ray data prior to (in the 6 months preceding) and after surgery (12 months postoperative) in patients operated on between 2009 and 2020 was performed (see Fig 1).

### Data

Data were collected by independent evaluators using a specific research algorithm corresponding to the French nomenclature of surgically performed acts. All variables of interest were subsequently extracted from the eligible charts. In addition, we contacted families through phone calls to document parent's feelings about their adolescent's QoL after surgery. Two independent evaluators (HB and EO) performed X-ray measurements to reduce ascertainment bias. In cases

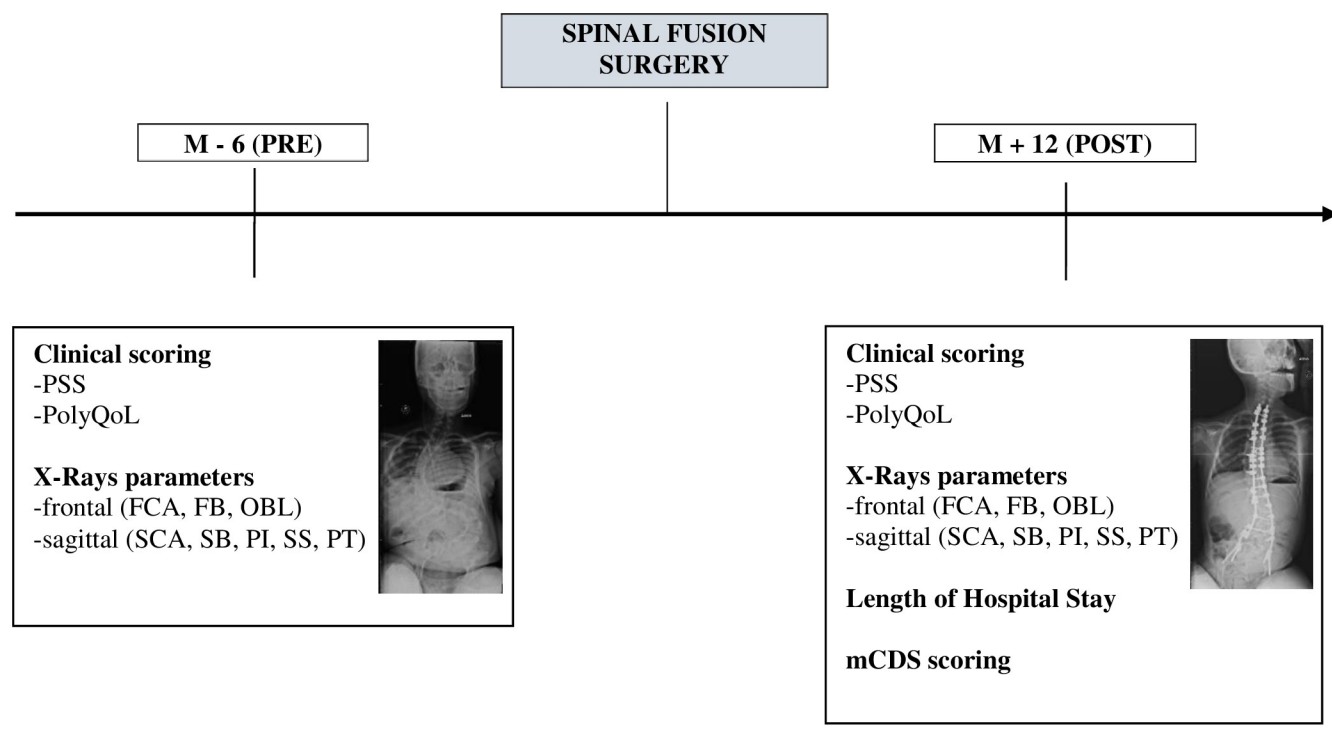

**Fig 1. Study design.** M Months; PSS Polyhandicap Severity Score; mCDS modified Clavien-Dindo-Sink classification; FCA Frontal Cobb Angle; FB Frontal Balance; OBL Pelvic Obliquity; SCA Sagittal Cobb Angle; SB Sagittal Balance; PI Pelvic Incidence; SS Sacral Slope; PT Pelvic Tilt.

of disagreement (>6° or >7° differences between two measurements, respectively in frontal and sagittal planes), the referring surgeon (BD) arbitrated the decision. For all outcomes, we only considered the earliest preoperative measurements and the latest postoperative ones.

Data extraction was completed from computerized charts by the first author (HB) who was not involved in the surgical procedure nor medical postoperative follow-up. Method for fulfilling evaluation form was as exhaustive as possible to limit risk of bias. Age at surgery, sex, care facility, preoperative FIM, past surgical history, PLH aetiology, type of physical disability, pain, body mass index (BMI), spontaneous posture, communication level, feeding strategies, and presence of medical devices were extracted. These data were compiled to rate the severity health status of each participant [19] according to the Polyhandicap Severity Scale (PSS). The PSS provides an accurate evaluation of health status regarding abilities, comorbidities and impairments, and assesses the level of global severity of the health status of persons with PLH. This scale ranges scores from 0 to 129, from less to higher levels of severity. A global PSS score was calculated as the sum of the two PSS subscores, respectively "comorbidities and impairments" (coPSS, ranging from 0 to 69) and "abilities" (abPSS, ranging from 0 to 60), according to the International Classification of Functioning, Disability and Health (ICF). Semi structured interviews (see Fig 2) were conducted with families to collect verbatim interviews. We also rated pre- and postoperative PolyQoL scores [20]. This validated short-scale is composed of two domains called health and social, ranging QoL scores from 0 (worst) to 100 (better).

X-ray parameters were extracted from pre- and postoperative X-rays (sagittal and frontal). A specific software program (Surgimap Spine® 2021, Nemaris, Inc.) was used for all spinal and pelvic measurements, as it offers semiautomatic procedures with good inter- and intraobserver reliability, even on instrumented spines. Frontal X-ray data included skeletal maturity

- What are the most significant elements during your children's hospitalization that you remember?

- What did the surgery do for or against your child?

- Would you say that the surgery had direct or indirect positive effects?

- Would you say that the surgery had direct or indirect negative effects?

- If you had to give a % of improvement/deterioration of the quality of life of your children comparing before and after the surgery what it would be?

- If you had to participate in the decision again, would you make the same one? Why?

**Fig 2. Semi structured interviews (phone calls).**

(Risser test), frontal Cobb angles (FCA), pelvic obliquity (OBL) and frontal balance (FB). Sagittal parameters included sagittal Cobb angles (SCA) and sagittal balance (SB). Using the same software, we measured pelvic incidence (PI), pelvic tilt (PT) and sacral slope (SS). Frontal and sagittal curves were added separately to obtain a "full curvature degree" for each participant, reflecting the amount of deformity.

Peri- and postoperative complications ranged according to the modified Clavien-Dindo-Sink classification [21] (mCDS). We collected blood loss data (loss of more than 1 blood volume), deep wound infections (DWI) and operative durations. We also calculated hospital lengths of stay, composed of the sum of paediatric surgery, intensive care unit (ICU), and rehabilitation department stays (days of hospitalization).

## Statistical analysis

We first checked if variables followed a normal distribution. Descriptive statistics were reported as means with standard deviation (SD) for quantitative variables and frequencies for qualitative variables (%). We used paired t tests for normally distributed variables for comparison of pre- and postoperative variables. Wilcoxon matched-pairs signed rank tests were used for nonparametric repartitions. Univariate analysis was conducted for postoperative PSS scores and PSS score variations and their related subscores, using Mann-Whitney tests for qualitative variables and Spearman correlation matrices for quantitative data (Prism Graph-Pad® software for Windows, version 5.03, San Diego California USA, www.graphpad.com). Significance threshold was set at $p < 0.05$.

## Results

### Participants

We registered 116 patients presenting with a neuromuscular disease or CP who underwent a surgical procedure for scoliosis. Among them, 30 patients (19 women, 11 men) met the definition of PLH (see flow chart—Fig 3).

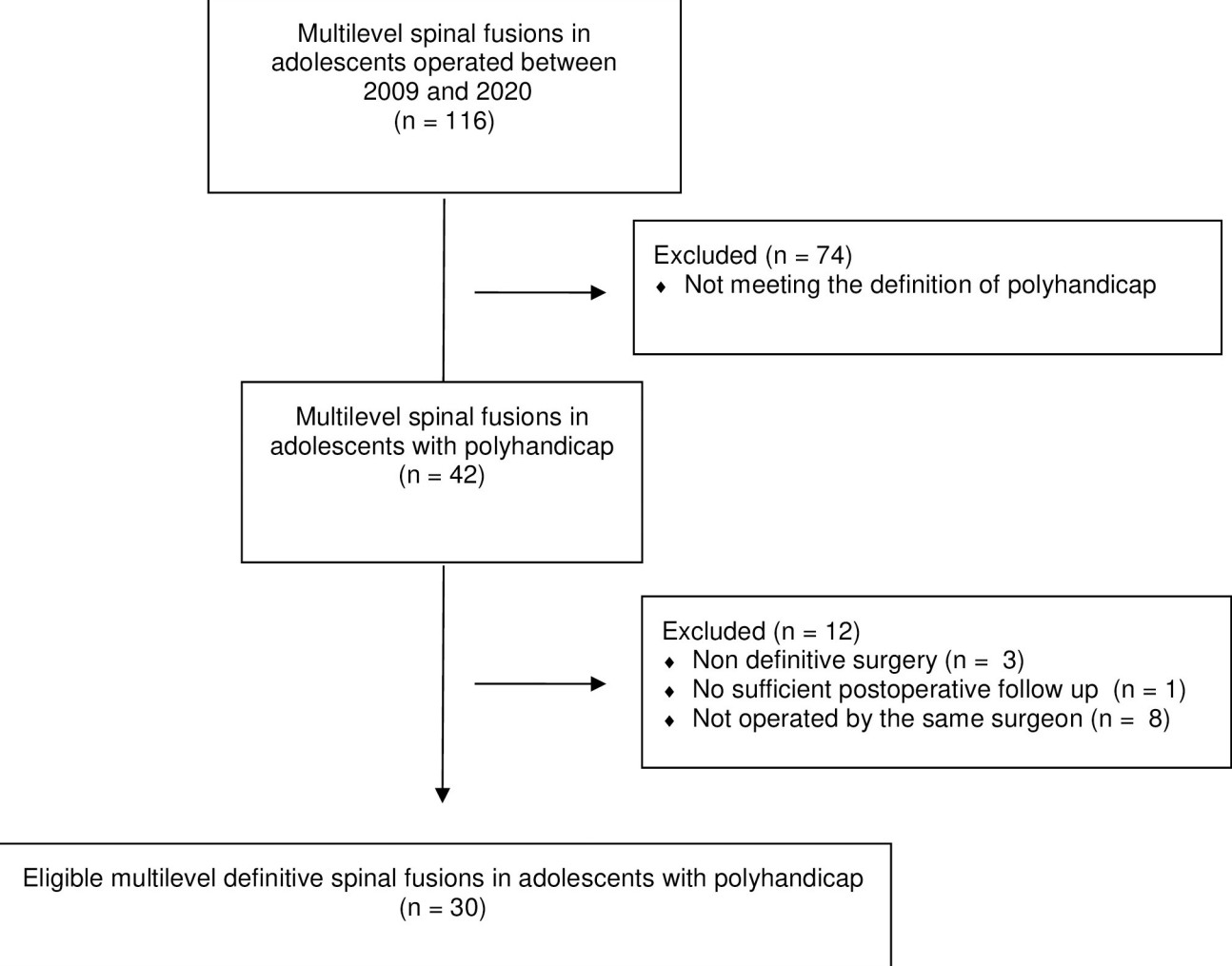

**Fig 3. Flow chart.**

### Descriptive data

The mean age at surgery was 15 (3) years old. The preoperative mean FIM was 22 (10). PLH aetiology was unknown for 8 adolescents and progressive for 6 adolescents. Twenty patients had undergone previous orthopaedic surgery, mainly multilevel tenotomy surgery (n = 9). The mean BMI before surgery was 17.4 (3.9) kg/m$^2$. Descriptive data are reported in Table 1.

The mean preoperative scores were 53 (17), 14 (5), and 39 (13) for PSS (/129), coPSS (/69) and abPSS (/60) respectively. For postoperative scores, we found a score of 49 (16) for PSS, a score of 12 (4) for coPSS and a score of 37 (13) for abPSS. Data were missing in 3 cases: 1 adolescent died during surgery and 2 medical records were insufficient for exhaustive PSS grading before and after surgery. Another adolescent died 7 days after surgery, but postoperative PSS scoring was possible. Seven caregivers agreed to answer semistructured interviews. The mean preoperative PolyQoL score was 64 (12). Mean postoperative PolyQoL score was 77 (10). Pre- and postoperative data, with related means of difference, are shown in Table 2. A scatter plot of individual preoperative versus postoperative PSS scores is provided in Fig 4.

**Table 1. Descriptive data.**

|  | n | % |
|---|---|---|
| **Mean age at surgery** (+/-sd) | 15.2 (+/-2.8) | NA |
| **Median age at surgery** [Interquartile Interval] | 14.7 [13.0;16.2] | NA |
| **Gender** |  |  |
| Female | 19 | 63% |
| Male | 11 | 37% |
| **Aetiology** |  |  |
| Unknown | 8 | 27% |
| Known | 22 | 73% |
| •Progressive | 6 | 20% |
| •Non progressive | 16 | 53% |
| **Past surgical history** |  |  |
| No previous surgery | 10 | 33% |
| At least one surgery | 20 | 67% |
| •Bone surgery | 5 | 17% |
| •Musculo tendinous surgery | 9 | 30% |
| •Combined surgery | 6 | 20% |
| **Medical devices** |  |  |
| Gastrostomy |  |  |
| •With | 14 | 47% |
| •Without | 16 | 53% |
| IT Baclofen pump |  |  |
| •With | 7 | 23% |
| •Without | 23 | 77% |
| **Type of care facility** |  |  |
| Home (only) | 2 | 7% |
| Rehabilitation facilities | 28 | 93% |

**Table 2. Variations in clinical and X-ray parameters, 6 months before and 12 months after spinal fusion.**

|  | Preoperative | Postoperative | Mean of difference | p values | CI |
|---|---|---|---|---|---|
| **PSS (/129)** | 56.5 [39.5;65.5] | 55 [39;62.5] | -2.8 (3.5) | **$3 . 10^{-4}$** | 1.5 to 4.2 |
| **coPSS (/69)** | 13.5 [11.3;19.5] | 11.5 [10;15] | -2.2 (2.9) | **$6 . 10^{-4}$** | 1.0 to 3.3 |
| **abPSS (/60)** | 45 [27;50] | 40 [26;49] | -0.7 (0.9) | **$9 . 10^{-4}$** | 0.3 to 1.0 |
| **PolyQoL (/100)** | 71 [62;81] | 85 [71;94] | 12.2 (12.7) | 0.09 | NA |
| **BMI (kg/m$^2$)** | 16.8 [15.1;19.4] | 18.3 [16.5;20.0] | 1.4 (1.5) | **$7 . 10^{-4}$** | -2.1 to -0.7 |
| **FCA (˚)** | 91 [62;106] | 39 [21;50] | -55.1 (37.3) | **$< 10^{-4}$** | 39.0 to 71.3 |
| **SCA (˚)** | 159 [96;178] | 95 [80;120] | -46.9 (51.6) | **$9 . 10^{-4}$** | 22.0 to 71.8 |
| **OBL (˚)** | 13 [7;18] | 7 [3;10] | -6.7 (6.1) | **$< 10^{-4}$** | 3.96 to 9.40 |
| **SS (˚)** | 36 [21;57] | 36 [31;57] | -5.4 (21.3) | 0.28 | -15.3 to 4.63 |
| **PT (˚)** | 27 [17;47] | 20 [14;35] | -7.8 (22.4) | **0.047** | 0.1 to 19.7 |

Medians with [IQR], means (sd), confidence intervals (CI: min to max). Significance threshold was set at $p < 0.05$.

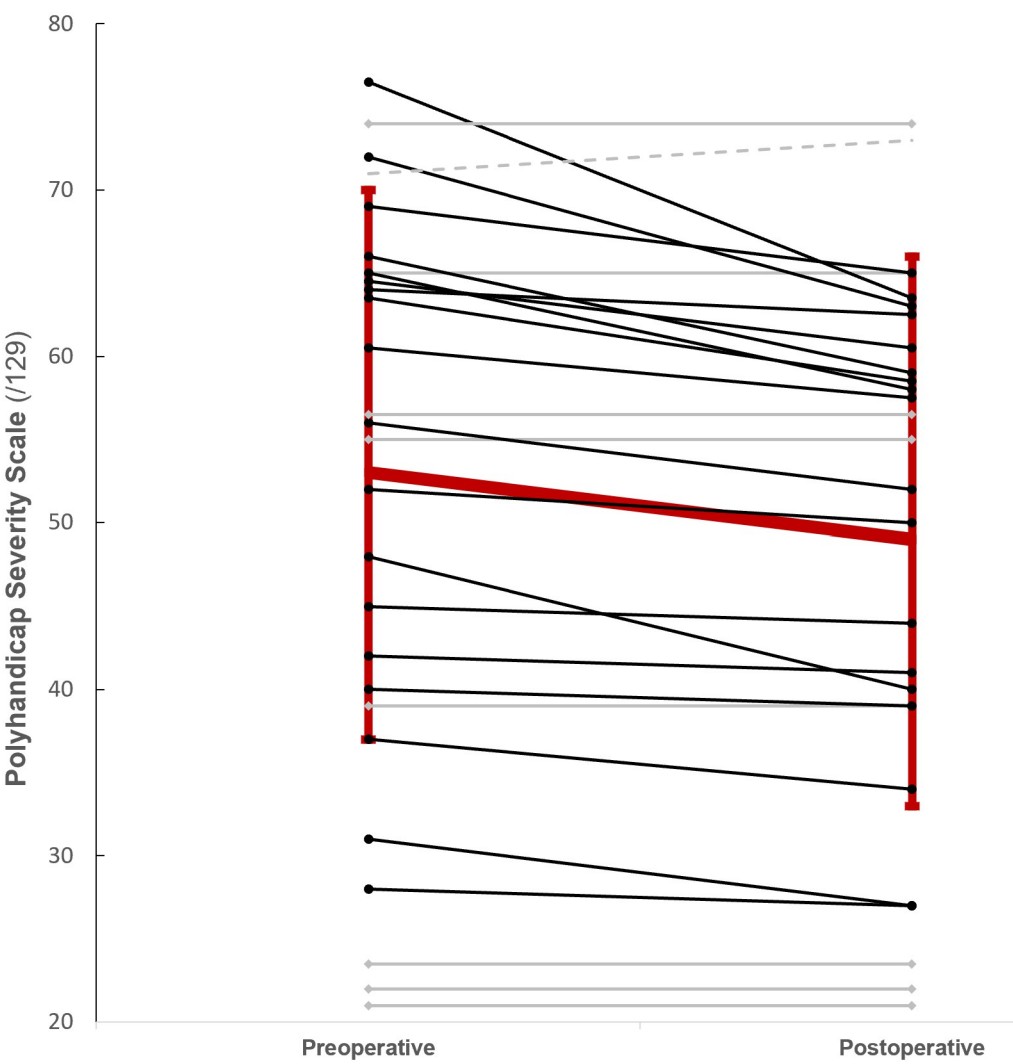

**Fig 4. Scatter plot of individual preoperative versus postoperative PSS scores.** Mean PSS evolution with standard error is represented by the large red solid line with error bar. PSS evolution i.e improvement (n = 18; dark solid line), stability (n = 8; grey solid line), worsening (n = 1; grey dashed line) are represented for each individual.

For complications, all the subjects presented at least 1 per- or postoperative complication. Among them, 9 were rated grade 1, and 10 were rated grade 4 according to the mCDS classification (Fig 5). More accurately, 6 presented with a deep wound infection (DWI), and 13 with a major blood loss. Neither neurologic, nor digestive or pancreatic complications were noted postoperatively in the patients. Finally, we found a mortality rate of 6.7% during follow-up: 1 death occurred during the surgical procedure, which was related to heart failure during anaesthesia, and 1 in the ICU due to multisystemic failure 7 days after surgery.

The mean hospital length of stay was 44.7 days (20 to 83 days), respectively comprising 3.6 (2.7) days (1 to 10) in the ICU, 14.3 (8.3) days (1 to 34) in the paediatric surgery department, and 26.9 (12.9) days (0 to 56) in the rehabilitation department.

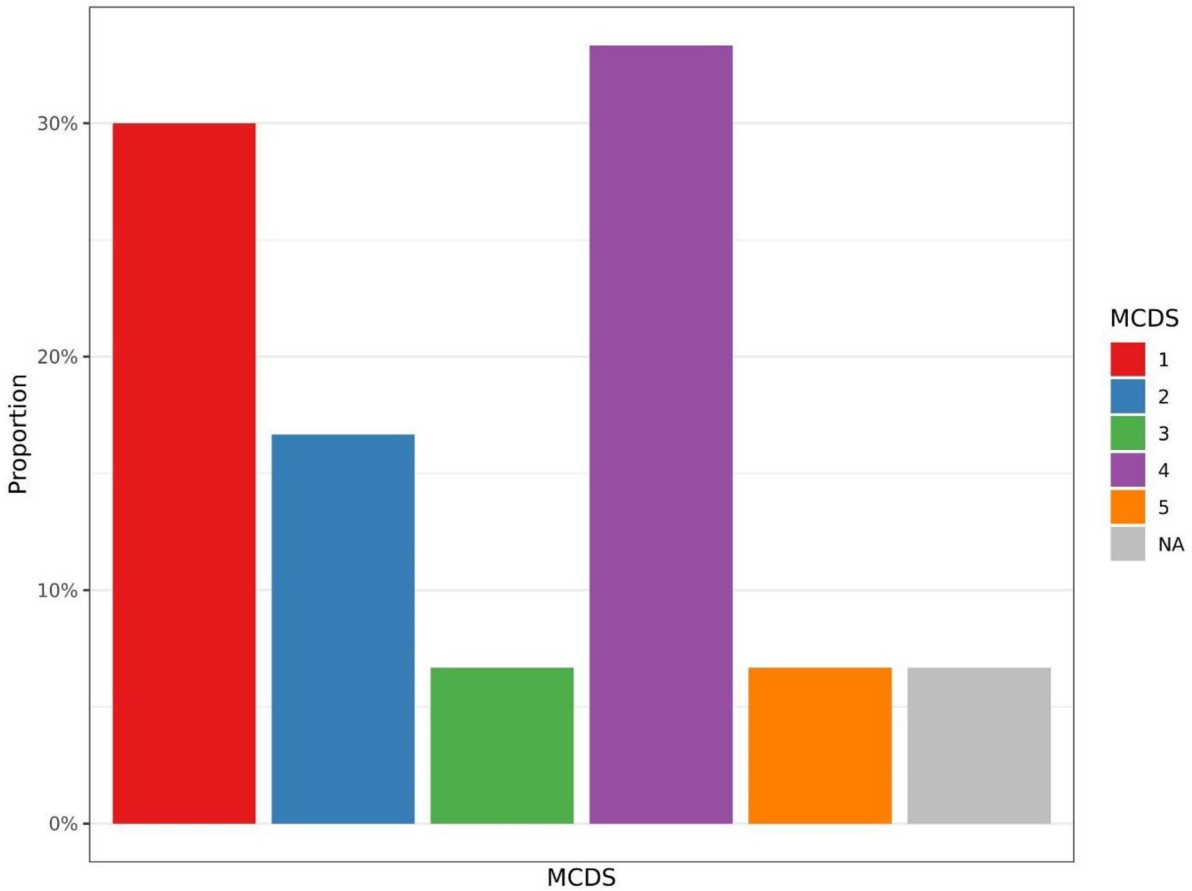

**Fig 5. Proportions (%) of complications as classified by mCDS scoring (minor to severe complications, from grade 1 to 5).** Grade 1 (n = 9): Wound leakage, constipation, nonsevere lung infection; Grade 2 (n = 5): Non severe dura mater breach, delayed wound healing; Grade 3 (n = 2): Deep wound infection, treated with second surgery + IV antibiotics; Grade 4 (n = 10): Any complication requiring ICU admission: Complex deep wound infection, renal failure, urinary sepsis, severe lung infection, delayed awakening after anaesthesia, central venous catheter infection; Grade 5 (n = 2): Per operative death or early postoperative death (within 7 days); NA (n = 2): Missing data.

### Analysis

We found significant variations (p < $10^{-3}$) between the pre- and postoperative assessments: -2.83 (3.49) for PSS, -2.17 (2.87) for coPSS, and -0.67 (0.92) for abPSS. Among 28 patients who survived surgery, we were able to fully analyse 27 charts. PSS scores worsened for 1 adolescent, PSS scores remained unchanged for 8 adolescents, and PSS scores improved for 18 adolescents. Pain was the first cause of improvement in coPSS assessment, found in 33% of all the adolescents (n = 9). This finding was followed by respiratory disability (n = 8), and digestive and skin disability improvements (each n = 7). Regarding behavioural disorders, improvements were found in 5 cases. For abPSS, 12 adolescents were improved, mainly for general, sitting, supine postures and feeding domains. None of the adolescents analysed presented with less ability after surgery, and 15 were stabilized (equal pre- and postscore).

Of all the families contacted, 7 agreed with answering semistructured interviews. We report here the most salient quotations: "surgery was inevitable" (family 11); "without this surgery, my daughter would have lost a sitting position" (family 21); and "PLH would have made the rest of her life very uncertain" (family 16).

For X-ray data, we highlighted statistically significant mean variations for FCA ($p < 10^{-4}$), SCA ($p < 10^{-3}$), OBL ($p < 10^{-4}$), PT ($p < 0.05$). We did not show any significant variations for SS ($p = 0.28$).

## Other analyses

In univariate analysis, we found that postoperative PSS scores were moderately correlated with preoperative FIM ($p < 10^{-3}$, $\rho_S$ = -0,61), BMI ($p = 0.02$, $\rho_S$ = -0,50), and OBL ($p = 0.03$, $\rho_S$ = -0,48) (see Table 3). Regarding PSS score variations, OBL was the unique preoperative factor we found to be correlated ($p < 0.05$, $\rho_S$ = -0,48). Specifically, coPSS score variations were correlated with preoperative FIM ($p = 0.04$, $\rho_S$ = 0,41), FCA ($p = 0.01$, $\rho_S$ = -0,53), and OBL ($p = 0.007$, $\rho$ = -0,56).

Baclofen pump implantation prior to spinal fusion was not associated with complication severity regarding mCDS ($p = 0,27$) or DWI ($p = 1.0$) (see Table 4). For hospital lengths of stay, the older the patients were, the shorter their ICU stay ($p < 10^{-3}$, $\rho_S$ = -0,69).

**Table 3. Univariate analysis of postoperative Polyhandicap Severity Scale scores.**

**(a)**

|  | Spearman ρ coefficient | p values |
|---|---|---|
| FIM | -0,61 | **< $10^{-3}$** |
| Year of surgery | -0,05 | 0,81 |
| Age at surgery | -0,05 | 0,80 |
| Risser score | 0,18 | 0,44 |
| preoperative BMI | -0,50 | **0,02** |
| preoperative FCA | 0,30 | 0,17 |
| preoperative OBL | 0,48 | **0,03** |
| preoperative FB | -0,08 | 0,71 |
| preoperative SCA | 0,36 | 0,15 |
| preoperative PI | 0,18 | 0,46 |
| preoperative SS | -0,26 | 0,28 |
| preoperative PT | 0,36 | 0,13 |
| preoperative SB | 0,27 | 0,28 |

**(b)**

|  | Groups | n | Mean Rank | Sum of Ranks | U | p values |
|---|---|---|---|---|---|---|
| Gastrostomy | With | 12 | 60.96 | 243 | 15 | < $10^{-3}$ |
|  | Without | 15 | 39.93 | 135 |  |  |
| IT Baclofen pump | With | 5 | 53 | 78.5 | 46.5 | 0.62 |
|  | Without | 22 | 48.43 | 299.5 |  |  |
| Sex | Female | 18 | 50.47 | 254.5 | 78.5 | 0.92 |
|  | Male | 9 | 46.89 | 123.5 |  |  |
| Aetiology | Progressive | 6 | 58.83 | 80 | 19 | 0.09 |
|  | Non progressive | 13 | 48.5 | 110 |  |  |
| Past surgical history | Yes | 17 | 53.26 | 269 | 54 | 0.13 |
|  | No | 10 | 42.5 | 109 |  |  |

(a) Spearman correlation matrix for quantitative preoperative outcomes (b) Mann-Whitney tests for qualitative preoperative outcomes. Significance threshold was set at $p < 0.05$.

**Table 4. Deep Wound Infection univariate analysis.**

| | | No DWI (n = 24) | DWI (n = 6) | p values |
|---|---|---|---|---|
| Gender* | Male | 6 (25%) | 5 (83%) | **0.016** |
| | Female | 18 (75%) | 1 (17%) | |
| IT Baclofen pump* | Without | 18 (75%) | 5 (83%) | 1 |
| | With | 6 (25%) | 1 (17%) | |
| Preoperative BMI | | 16.8 [15.5;18.9] | 16.5 [15.1;19.3] | 1 |
| Preoperative PSS | | 55 [40;65] | 65 [58;65] | 0.43 |
| Age at surgery | | 15 [12.9;17.3] | 14.7 [14.3;14.8] | 0.98 |

Medians with [IQR] (Mann-Whitney tests) and proportions (Fisher exact tests*). Significance threshold was set at p < 0.05.

## Discussion

In the present study, we aimed to describe comorbidities, impairments, activity limitations and QoL before and after spinal fusions in adolescents with PLH. We hypothesized that these surgeries lowered the severity health status and conferred clinical benefits. Our hypothesis was confirmed as PSS, coPSS and abPSS scores were significantly improved. We have nevertheless highlighted a 100% rate of complications and 2 deaths, testifying to the complexity of these surgeries.

Comorbidities, impairments, and activity limitations were evaluated, as outlined above, with the PSS. This scale allows us to quantify the severity health status of patients with PLH, providing a new ICF-standardized and exhaustive approach for clinical assessments. We believe in that better preoperative clinical specifications could help in identifying patients who may better tolerate and benefit from spinal fusions. Scoliosis surgery seems to represent a strong contribution to the long-term management of health in persons with PLH [2]. The results of our study demonstrate that spinal fusion surgery in adolescents with PLH leads to a reduction in comorbidities (respiratory, digestive, skin, behavioural, pain, etc.) and seems to allow a slight improvement in neurodevelopmental status.

Moreover in this study, we could interview seven of all families through semistructured questionnaires: witness families reported that spinal fusions conferred a global improvement in QoL. The parents' statements were mostly positive: during the preoperative period, they had difficulty accepting the vital risk of spinal surgery, but in retrospect, they recognized 1) that the surgery was necessary, and 2) that their child's health (breathing, posture. . .) had improved even when peroperative and immediate postoperative complications surrounded. Moreover, we used a validated tool called the PolyQoL questionnaire to objectively document QoL. We did not show any significant improvement regarding PolyQoL scores, but these results must be taken with caution, likely due to our small sample of answers.

Parents and caregivers were particularly attentive to the experience before and during the surgical period and reported having psychologically benefited from preoperative comprehensive explanations. Half of them verbalized the "inevitability" of this surgery. As highlighted by Adams et al. [22], agreements on the goals of surgery between surgeons and caregivers are important. While surgeons and physicians tend to give top priority to sitting considerations, caregivers put head control and physical appearance first. The expected benefits seem to remain superior to the risk of complications, but the therapeutic strategy requires an individualized evaluation rigorously explained to caregivers of PLH adolescents [22,23]. The question of long-term clinical benefits remains important, as follow-up is rarely conducted for more than 1 year [24].

Families are often afraid when scoliosis surgery is considered for their adolescent [25]. The postoperative complication rate (100%) and perioperative mortality rate (6,7%) must be included in the decision and balanced with improvements in severity health status. In our study, the DWI rate was 20%, which is almost twice the rate as previous studies reported in CP [26] (but CP adolescents present a less severe health status than PLH adolescents we analysed). Nutritional status of patients with PLH could be questioned to explain the difference.

Geometric spine readjustment could be one of the reasons for pain alleviation by costo-pelvic impingement prevention and sitting comfort improvement [27]. We observed a reduction in bed sore frequency and gluteal erosions after surgery. We confirmed a significant improvement in BMI after surgery (mean variation of +1,37 kg/m$^2$). As presumed in some studies, straightening the thoracolumbar spine could increase the abdominal space, allowing better peristalsis and less regurgitation [28]. However, severe postoperative pancreatitis cases have been reported after extensive arthrodesis, but we did not report any case in our population.

Regarding medical devices, we observed that IT baclofen pumps did not increase the complication rate, which is consistent with previous studies [29]. Similarly, gastrostomies were not associated with a major complication occurrence in our PLH patients, unlike CP GMFCS V patients in whom an increased risk has been shown previously [30]. Except for DWI, severe complications were mostly respiratory. Lung developmental defects, impaired neurologic command over added lung restrictive syndromes and postoperative intensive care with lung infection risks can partially explain our findings [6]. Preoperative breathing hyper insufflation strategies could be worthwhile as they demonstrated interest in spinal fusions in children with neuromuscular flaccid scoliosis [31]. As preoperative primary care visits led to lower costs and shorter hospitalizations in complex scoliosis surgeries, we support the idea of efficient preoperative rehabilitation programs before spinal fusions. Rehabilitation modalities could be determined according to the domains impacted in the preoperative PSS scoring.

Cobb corrections were 58% for FCA and 30% for SCA. Spinal fusions in adolescents with PLH provide comparable results to those found in GMFCS IV and V adolescents with CP (50 to 68% as reported). A mean PT variation of -7.8˚ (22.4) was observed, corresponding to a significant reduction in pelvis back-tilting. The mean correction for OBL was 47%. This X-ray parameter was correlated with the postoperative PSS score and PSS variations induced by surgery.

## Strengths and limitations

Our study demonstrates for the first time, that spinal fusions reduce the global severity of adolescents with polyhandicap. However, it contains several limitations: our study was monocentric and retrospective data collection may have provided methodological bias. Broadening inclusions to various specialized centers could be of interest in a prospective approach. Longer follow-up (2–5 years) would enable us to assess the long-term effect of spinal fusion surgery on the severity health status of patients with PLH, thus further prospective studies with a longer follow-up are needed (therefore, the present one-year follow-up evaluation should not be considered as the final evaluation). By increasing the duration of follow-up and through the methodical use of standardized tools such as PSS [19] and PolyQoL [20], establishing accurate and repeatable ICF-based descriptions in patients with PLH appears to be easier. An increased sample size could also help in analysing predictive factors of clinical improvement after spinal fusions, keeping in mind the potential confounding factors we highlighted.

One of the main issues when performing spinal fusions in patients with PLH is the assessment of predicted benefits. This study aimed to support the rationale for spinal fusion, acknowledging the already known complication rates and risks in PLH adolescents.

Considering the severity of the health status evaluation of these patients and the expected improvement of the global health status and comfort after spine surgery for scoliosis, each case requires careful consideration and evaluation. We support that, under the condition of acceptable preoperative health status, the indication for spinal fusion in that population could be relevant and reasonable. While retrospective evaluation is questionable, the rigorous method used during data collection should limit this bias. The PolyQoL questionnaire has been supported by the verbatim collected in a part of the population, and obviously the good health status of the patients at final follow-up confirms our hypothesis.

## Conclusions

To our knowledge, this is the first study demonstrating that spinal fusions confer a significant improvement in abilities, comorbidities and impairment scores in a population of adolescents with PLH one year after surgery. By using PSS and PolyQoL scores, ICF-standardized measurement tools, we emphasize the validity of these surgical practices, when indicated. Prospective longitudinal studies could be helpful by identifying preoperative relevant parameters testifying altered QoL and deteriorated health status, pointing out the interest of surgery when PLH adolescents present with scoliosis with Cobb angle >50˚. Practitioners should be informed of the importance of performing spinal fusion surgery in early adolescence of PLH patients in order to improve their global health status. While spinal fusions showed some improvement in the severity health status, one should be aware of the extremely high rate of perioperative complications fortunately treatable but often requiring readmission, making multidisciplinary expertise mandatory

## Supporting information

**S1 Data.**
(XLSX)

## Acknowledgments

We extend our thanks to the patients and families for their participation in this study. We acknowledge the support of Diana Rimaud (PhD), for statistical analysis supervision.

## Author Contributions

**Conceptualization:** Hugo Bessaguet, Marie-Christine Rousseau, Vincent Gautheron, Bruno Dohin.

**Data curation:** Hugo Bessaguet, Bruno Dohin.

**Formal analysis:** Hugo Bessaguet, Marie-Christine Rousseau, Vincent Gautheron.

**Investigation:** Hugo Bessaguet, Bruno Dohin.

**Methodology:** Hugo Bessaguet, Marie-Christine Rousseau, Etienne Ojardias.

**Project administration:** Hugo Bessaguet, Marie-Christine Rousseau, Vincent Gautheron, Bruno Dohin.

**Resources:** Hugo Bessaguet.

**Software:** Hugo Bessaguet, Etienne Ojardias.

**Supervision:** Marie-Christine Rousseau, Vincent Gautheron, Etienne Ojardias, Bruno Dohin.

**Validation:** Hugo Bessaguet, Marie-Christine Rousseau, Vincent Gautheron.

**Visualization:** Hugo Bessaguet.

**Writing – original draft:** Hugo Bessaguet.

**Writing – review & editing:** Hugo Bessaguet, Marie-Christine Rousseau, Vincent Gautheron, Etienne Ojardias, Bruno Dohin.

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
