## [Editor Report · Decision Letter 0]

23 Jan 2023

PONE-D-23-01480Impact of spinal fusion on severity health status in scoliotic adolescents with polyhandicapPLOS ONE

Dear Dr. Bessaguet,

Thank you for submitting your manuscript to PLOS ONE. After careful consideration, we feel that it has merit but does not fully meet PLOS ONE’s publication criteria as it currently stands. Therefore, we invite you to submit a revised version of the manuscript that addresses the points raised during the review process.

We look forward to receiving your revised manuscript.

Kind regards,

Malgorzata Wojcik, Ph.D

Academic Editor

PLOS ONE

Journal Requirements:

" The funders had no role in study design, data collection and analysis, decision to publish, or preparation of the manuscript."

Additional Editor Comments:

Thank you for the opportunity to review the article, the authors have tackled an important and difficult topic.

1. The abstract says: We included scoliotic adolescents 37 with polyhandicap who underwent spinal fusion surgeries from 2009 to 2020. Was the surgical technique the same from 2009 to 2020?

2. Hospital stay of 45 days - add important information how were patients treated after surgery? Physiotherapy? Corset treatment? Drug treatment? Other treatment?

3. Did the authors take into account the recommendations of SOSORT?

4. Could the authors elaborate on this sentence?Two independent evaluators performed X-ray measurements to reduce ascertainment bias. In cases of disagreement, the referring surgeon arbitrated the decision.

5. Statistical analysis - the authors did not state to what level of significance they refer their results, moreover, the tables in which there are p-values should also include the level of significance to which the authors refer.

6. Why are descriptive statistics presented as means with standard deviation (SD) for quantitative variables and frequencies for qualitative variables (%)? Why are the median and 1st and 3rd quartile values not presented?

7. Prism GraphPad software - please add more information about the software, there is no reference, this should also be completed.

8. Table 1 - the authors should add the age of the subjects.

9. Results - in addition to tables, the authors should add figures that would enrich the simple statistical methods used.

10. The authors should add what the limitations were in this study.

11. Conclusion - state what practical implications arise from this study, how they can be used - applied.

12. Please add current references 2013-2023 and, please add reference to Prism GraphPad software.

---

## [Author Response · Author response to Decision Letter 0]

28 Mar 2023

Dear Editor, thank you for all the comments. Here is a detailed point-by-point response corresponding to the suggested changes made to the manuscript:

To Academic Editor - March 28th 

1. "Thank you for providing the ethics documentation as reply to our previous request.

Please include an English translation of the ethics committee approval letter as an "Other" file. Please note that no officially translated version of the documents is required, but that the authors themselves can translate the ethics documents for this purpose.

Thank you for your attention. We look forward to hearing from you."

Authors

We thank the Academic Editor for this request. We uploaded as required an English translation of the ethics committee approval letter as an “Other” file (Ethics Statement – English version).

2. 

2.1) Please provide the specific name of the ethics Committee that approved your study

Authors

The Ethics Committee which approved our study is “Terre d’Ethique” Research Committee :

- Institutional Review Board : IORG0007394

- Study label : IRBN492021/CHUSTE (Saint-Etienne University Hospital Centre)

We detailed this information in the Method section (page 6, line 146) and added the IRB proof with this submission (Other “Ethics Statement”)

2.2) In the ethics statement in the Methods, you have specified that verbal consent was obtained. Please provide additional details regarding how this consent was documented and witnessed, and state whether this was approved by the IRB

Authors

Thank you for this suggestion. We notice that the written information was submitted and validated to the precited Ethics Committee before caregivers diffusion. We added required precisions on the dedicated paragraph :

“All familial caregivers received written information about the study, according to French law, for retrospective studies. A written consent was collected from all caregivers” in the Methods section (page 6, lines 148-149).

To Reviewers

1. The abstract says: We included scoliotic adolescents 27 with polyhandicap who underwent spinal fusion surgeries from 2009 to 2020. Was the surgical technique the same from 2009 to 2020?

Authors

To consider this comment we changed the sentence (page 2, line 36) “We included scoliotic adolescents with polyhandicap who underwent spinal fusion surgeries from 2009 to 2020” by: “We included between 2009 to 2020, 30 scoliotic adolescents with polyhandicap who underwent spinal fusion performed with the same surgical technique and the same surgeon.”

2. Hospital stay of 45 days - add important information how were patients treated after surgery? Physiotherapy? Corset treatment? Drug treatment? Other treatment?

Authors

Each patient received personalized post-operative rehabilitation consisting in medication against spasticity and pain as needed, physiotherapy, and moltened seat confection. We added these notions to the original sentence (page 2, line 47): The mean hospital length of stay was 45 days. During postoperative period, patients received a personalized post-operative rehabilitation procedure with spasticity and pain treatments, physiotherapy, and verticalization (wheelchair sitting and positioning devices such as contoured seat intended to increase postural stability).

3. Did the authors take into account the recommendations of SOSORT?

Authors 

We thank the reviewer for this interesting suggestion. Nevertheless, the SOSORT recommendations are for conservative treatment of idiopathic scoliosis. The present study targets non conservative treatment (spinal fusion surgery) for scoliotic adolescents with polyhandicap and could not include these recommendations. Therefore, the present manuscript doesn't add any changes related to the SOSORT recommendations.

4. Could the authors elaborate on this sentence? Two independent evaluators performed X-ray measurements to reduce ascertainment bias. In cases of disagreement, the referring surgeon arbitrated the decision.

Authors

Two independent evaluators (HB and EO) performed independent X-ray measurements using the semiautomatic procedures of the Surgimap Spine® software. In cases of disagreement (>6° or >7° differences between two measurements, respectively in frontal and sagittal planes as described by “Wu W, Liang J, Du Y, Tan X, Xiang X, Wang W, et al. Reliability and reproducibility analysis of the Cobb angle and assessing sagittal plane by computer-assisted and manual measurement tools. BMC Musculoskelet Disord. 2014 Dec;15(1):33.”), the referring surgeon (BD) was charged to arbitrate the decision.

To clarify the procedure, we added the following informations (page 7, line 164): “Two independent evaluators (HB and EO) performed X-ray measurements to reduce ascertainment bias. In cases of disagreement (>6° or >7° differences between two measurements, respectively in frontal and sagittal planes), the referring surgeon (BD) arbitrated the decision.”

5. Statistical analysis - the authors did not state to what level of significance they refer their results, moreover, the tables in which there are p-values should also include the level of significance to which the authors refer.

Authors

Significance threshold was set at p < 0.05. We added the significance threshold both in the “Statistical analysis” section (page 9, line 211) and below in the Tables (2., 3. and 4.)

6. Why are descriptive statistics presented as means with standard deviation (SD) for quantitative variables and frequencies for qualitative variables (%)? Why are the median and 1st and 3rd quartile values not presented?

Authors

We thank the reviewer to offer us the opportunity to clarify our results.

Table 1: we added the mean and median ages at surgery of our subjects (point 8.) and better described our population (number of subjects = n and their related proportions = %)

Table 2: All data presented are quantitative data (clinical scores, body mass index, angles). However, to make it clearer, we made the following corrections, expressing as recommended our data as medians [1st quartile – 3rd quartile]

7. Prism GraphPad software - please add more information about the software, there is no reference, this should also be completed.

Authors

We added more accurate information in the method section about the software version we used (page 9, line 211) : Prism GraphPad® software for Windows, version 5.03, San Diego California USA, www.graphpad.com. This software was registered to CHUSE and Saint-Etienne university under the license GPW5-865491-RAG-9037.

8. Table 1 - the authors should add the age of the subjects.

Authors

We added the age of the subjects in Table 1(mean/median ages with sd / 1st and 3rd quartile - page 10, line 231)

9. Results - in addition to tables, the authors should add figures that would enrich the simple statistical methods used.

Authors

As a way to enrich our statistical methods, we added a study description figure (page7, line 157) as presented above (Figure 1. Study design). We pay attention to use Preflight Analysis and Conversion Engine (PACE) digital diagnostic tool before submitting it (file format .tif).

Fig 1. Study design. M Months; PSS Polyhandicap Severity Score; mCDS modified Clavien-Dindo-Sink classification; FCA Frontal Cobb Angle; FB Frontal Balance; OBL Pelvic Obliquity; SCA Sagittal Cobb Angle; SB Sagittal Balance; PI Pelvic Incidence; SS Sacral Slope; PT Pelvic Tilt.

We took care to modify the format of the 3 others figures, the numbers in manuscript body, and added respective legends in text conferring to figure caption formatting guidelines.

10. The authors should add what the limitations were in this study.

Authors

To consider the reviewer’s comment, we added a « Strengths and Limitations” paragraph at the end of the discussion section paragraph) (page 19, line 396). “Our study demonstrates for the first time, that spinal fusions reduce the global severity of adolescents with polyhandicap. However, it contains several limitations: our study was monocentric and retrospective data collection may have provided methodological bias.”

11. Conclusion - state what practical implications arise from this study, how they can be used - applied.

Authors

To highlight the implications for clinical practice we added the sentence (page 20, line 415): “Practitioners should be informed of the importance of performing spinal fusion surgery in early adolescence of PLH patients in order to improve their global health status”.

12. Please add current references 2013-2023 and, please add reference to Prism GraphPad software.

Authors

Thank you for this comment. We added Prism GraphPad® software specifications in point 7. (Windows, version 5.03, San Diego California USA, www.graphpad.com). We have updated the reference 19. Hamouda I, Rousseau MC, Aim MA, Anzola AB, Loundou A, De Villemeur TB, et al. Development and initial validation of the quality of life questionnaire for persons with polyhandicap (PolyQoL). Ann Phys Rehabil Med. févr 2023;66(1):101672. which is the most recent indexed citation dealing with polyhandicap to our knowledge.

---

## [Decision Letter · Decision Letter 1]

19 Dec 2023

PONE-D-23-01480R1Impact of spinal fusion on severity health status in scoliotic adolescents with polyhandicapPLOS ONE

Dear Dr. Bessaguet,

Thank you for submitting your manuscript to PLOS ONE. After careful consideration, we feel that it has merit but does not fully meet PLOS ONE’s publication criteria as it currently stands. Therefore, we invite you to submit a revised version of the manuscript that addresses the points raised during the review process.

We look forward to receiving your revised manuscript.

Kind regards,

Kentaro Yamada, M.D., Ph.D.

Academic Editor

PLOS ONE

Additional Editor Comments:

Thank you for your contribution to this journal. We apologize for the considerable time that has elapsed since you once received an appropriate Revision to the reviewer's comment. In fact, due to an error in our journal system, we sent you a first Revise letter without reaching the required number of reviewers determined by our journal policy.

We have now done additional reviewers and have received comments that major revisions may be necessary.

We apologize for the inconvenience, but would you please consider revising the additional reviewer's comments again?

We again apologize for the time it will take in a hurry to publication.

Reviewers' comments:

Reviewer's Responses to Questions

**Comments to the Author**

1. If the authors have adequately addressed your comments raised in a previous round of review and you feel that this manuscript is now acceptable for publication, you may indicate that here to bypass the “Comments to the Author” section, enter your conflict of interest statement in the “Confidential to Editor” section, and submit your "Accept" recommendation.

Reviewer #1: (No Response)

Reviewer #2: (No Response)

2. Is the manuscript technically sound, and do the data support the conclusions?

Reviewer #1: No

Reviewer #2: Partly

3. Has the statistical analysis been performed appropriately and rigorously? 

Reviewer #1: Yes

Reviewer #2: Yes

4. Have the authors made all data underlying the findings in their manuscript fully available?

Reviewer #1: Yes

Reviewer #2: No

5. Is the manuscript presented in an intelligible fashion and written in standard English?

Reviewer #1: Yes

Reviewer #2: Yes

6. Review Comments to the Author

Reviewer #1: Thank you for the submission of your research work to the journal. However, I have few concerns.

I do not agree with the authors stating severe scoliosis as Cobb angle more than 50 degrees. Severe scoliosis is usually defined when Cobb angle of more than 80 - 100 degrees.

To evaluate surgical outcomes, a minimum 2-year follow-up should be used rather than 1-year follow-up.

As mentioned by the authors, the preoperative assessment was done retrospectively by referring to the medical notes and this will create a major bias.

The sample population is heterogenous, perhaps should only focus on 1 diagnosis e.g. cerebral palsy. The sample size is low. As highlighted by the authors, a multicenter study should be conducted instead.

All patients had complications (100%), and there were 2 deaths out of 30 patients who underwent surgery (6.7% risk of death).

The authors did not mention if there were any neurologic deficits, venous thromboembolism, rate of unplanned return to OR. These are some important complications that are important to spine surgeons.

While surgical intervention showed some improvement in the severity health status in this group of patients, one should be aware of the extremely high rate of perioperative complications.

Reviewer #2: The study investigates the impact of spinal fusion surgeries on the severity of health status in adolescents with polyhandicap and scoliosis. It is a monocentric retrospective observational study, examining patients who underwent spinal fusion surgeries at the University Hospital Centre of Saint-Etienne, France, from 2009 to 2020. The primary outcome measured was the variation in the Polyhandicap Severity Scale (PSS) score post-surgery, with secondary outcomes including variations in PSS subscores, quality of life scores, X-ray parameters, and surgical complication rates and lengths of stay.

Data for the study were collected using a specific research algorithm and independently reviewed. Variables of interest were extracted from eligible charts, and families were contacted to document their perspectives on the adolescents' quality of life post-surgery. Statistical analysis included checking for normal distribution of variables, using paired t-tests, Wilcoxon matched-pairs signed rank tests, and univariate analysis with Mann-Whitney tests and Spearman correlation matrices.

The study included 30 patients (19 women and 11 men) who met the definition of polyhandicap. Significant improvements were observed between pre- and postoperative PSS scores, particularly in aspects like pain, respiratory, digestive, and skin disabilities. However, there was a 100% rate of complications, with at least one complication per patient and a mortality rate of 6.7% during follow-up.

While the study offers valuable insights into the effects of spinal fusion on adolescents with polyhandicap, addressing these points could enhance the depth and applicability of the findings:

1. Research design: The retrospective observational nature of the study is appropriate for the research question, but the existence of confounding factors, such as patient demographic information, consistency of the surgery, post-surgery treatments etc., limits the generalizability of the findings. Providing information or comment on this would strengthen the study's credibility and validity.

2. Statistical analysis: While the authors provide mean and standard deviations, the significance levels of the analyses should be clearly indicated. Additionally, a graphical representation of each patient's pre- and post-surgery scores would provide a clearer understanding of the data distribution, highlighting any potential skewness or outliers.

3. Implications and limitations of the study: I suggest the authors mentioning the practical implications and limitations of the study, which is important for understanding the scope and applicability of the findings.

7. PLOS authors have the option to publish the peer review history of their article (what does this mean?). If published, this will include your full peer review and any attached files.

Reviewer #1: No

Reviewer #2: No

---

## [Author Response · Author response to Decision Letter 1]

22 Jan 2024

Dear Editor, Dear Reviewers,

We would like to thank you for all the comments, which have helped us to improve the quality of our article. Here are detailed point-to-point responses corresponding to the suggested changes made to our manuscript entitled “Impact of spinal fusion on severity health status in scoliotic adolescents with polyhandicap”. All data underlying the findings of our manuscript is fully available in Supporting Data (dataSPH.xlsx).

We hope you will find this third version satisfactory and that it will be suitable for publication in your journal. Thank you for your consideration.

Sincerely, 

HB

---

## [Decision Letter · Decision Letter 2]

21 Feb 2024

Impact of spinal fusion on severity health status in scoliotic adolescents with polyhandicap

PONE-D-23-01480R2

Dear Dr. Bessaguet,

We’re pleased to inform you that your manuscript has been judged scientifically suitable for publication and will be formally accepted for publication once it meets all outstanding technical requirements.

Kind regards,

Kentaro Yamada, M.D., Ph.D.

Academic Editor

PLOS ONE

Reviewers' comments:

Reviewer #2: The authors addressed my comments, especially the addition of graphical representations of pre- and post-surgery scores, as it provides a clear and intuitive understanding of the data distribution and individual patient outcomes.

While the limitations inherent to the retrospective nature of the study and the relatively small sample size remain, the acknowledgment of these issues and the steps taken to mitigate their impact are noted. The data presented, despite the highlighted limitations and the complexity of the patient population, provide insights into the potential benefits and risks associated with spinal fusion surgeries in this group.

In light of the above, and considering the partial support provided by the data for the study's conclusions, I support the acceptance of this manuscript for publication.

---

## [Editor Report · Acceptance letter]

26 Feb 2024

PONE-D-23-01480R2 

PLOS ONE

Dear Dr. Bessaguet, 

I'm pleased to inform you that your manuscript has been deemed suitable for publication in PLOS ONE. Congratulations! Your manuscript is now being handed over to our production team.

Kind regards, 

on behalf of

Dr. Kentaro Yamada 

Academic Editor

PLOS ONE